# Precious Metal-Free CoP Nanorod Electrocatalyst as an Effective Bifunctional Oxygen Electrode for Anion Exchange Membrane-Unitized Regenerative Fuel Cells

**Palanisamy Rajkumar [1,†]**, **Md. Masud Rana [1]**, **Beom-Soo Kang [1]**, **Ho-Jung Sun [2]**, **Gyungse Park [3]**, **So-Yeon Kim [4]**, **Hong-Ki Lee [5]** and **Joongpyo Shim [1,*]**

[1]   Department of Chemical Engineering, Kunsan National University, Jeonbuk 54150, Republic of Korea; rajphysics@yahoo.com (P.R.); masudrana.iu21@gmail.com (M.M.R.); nummber1@naver.com (B.-S.K.)

[2]   Department of Material Science and Engineering, Kunsan National University, Jeonbuk 54150, Republic of Korea; hjsun@kunsan.ac.kr

[3]   Department of Chemistry, Kunsan National University, Jeonbuk 54150, Republic of Korea; parkg@kunsan.ac.kr

[4]   Department of Chemical Engineering Education & Graduate School of Energy Science and Technology, Chungnam National University, Daejeon 34134, Republic of Korea; kimsy@cnu.ac.kr

[5]   Fuel Cell Regional Innovation Center, Woosuk University, Jeonbuk 55315, Republic of Korea; hongkil@woosuk.ac.kr

\*   Correspondence: jpshim@kunsan.ac.kr

†   Current address: Department of Mechanical Engineering, Yeungnam University, Gyeongbuk 38541, Republic of Korea.

**Abstract:** In this study, noble metal-free Co(OH)F and CoP nanorod electrocatalysts were prepared and explored as bifunctional oxygen electrodes (BOE) in anion exchange membrane-unitized regenerative fuel cells (AEM-URFCs). A CoP nanorod was synthesized from Co(OH)F via the hydrothermal treatment of cobalt nitrate, ammonium fluoride, and urea, followed by phosphorization. The crystal structures, surface morphologies, pore distributions, and elemental statuses of the obtained catalysts were analyzed to identify the changes caused by the incorporation of fluorine and phosphorus. The presence of F and P was confirmed through EDS and XPS analyses, respectively. Using these catalysts, the AEM-based URFCs were operated with hydrogen and oxygen in the fuel cell mode and pure water in the electrolysis mode. In addition, the electrocatalytic activities of the catalysts were evaluated using cyclic voltammetry and electrochemical impedance spectroscopy. In the AEM-URFC test, the CoP catalyst in the BOE delivered the best performance in the fuel cell mode (105 mA cm$^{-2}$ at 0.3 V), and Co(OH)F was suitable for the water electrolyzer mode (30 mA cm$^{-2}$ at 2.0 V). CoP and Co(OH)F exhibited higher round trip efficiency (RTE) and power densities than the conventional $Co_3O_4$ catalyst.

**Keywords:** anion exchange membrane; fuel cells; electrocatalyst; oxygen electrode; electrolyzer

## 1. Introduction

Unitized regenerative fuel cells (URFCs) are widely recognized as effective and affordable energy conversion and storage systems, particularly when integrated with renewable resources [1]. The URFC is a hybrid device that can function as both a fuel cell and electrolyzer in a single device. URFCs, which combine fuel cells and electrolyzers, are emerging as promising power sources with high energy density for special applications that simultaneously produce clean hydrogen and oxygen in the electrolyzer mode and electricity in the fuel cell mode. Unlike conventional energy storage systems, URFCs are energy-efficient, inexpensive, simple to design, and environmentally friendly [2–5]. In the last decade, proton exchange membrane (PEM)-based URFC technology has proven to be very efficient owing to its superior cell performance and high round-trip efficiency [6]. Platinum-group

metal (PGM) electrocatalysts are widely used for oxygen electrode reactions in acidic media, but their scarcity and high price limit their scalability [1]. This has prompted researchers to explore the possibility of anion exchange membrane-unitized regenerative fuel cells (AEM-URFCs) with improved reactivity and reduced electrocatalyst costs and stack components (membranes, bipolar plates, air looping, cooling, etc.) [7]. Furthermore, concentrated KOH can be replaced with distilled water, making it possible to replace Nafion with a low-cost hydrocarbon membrane [8].

There have been great challenges regarding conventional PGM-based catalysts, such as Pt and Ir/Ru (oxides), including limited resources and high costs, thus reducing their market competitiveness. Developing non-noble metal bifunctional catalysts that perform well in alkaline media is crucial to achieving ideal URFC performance. Accordingly, transition-metal-based bifunctional catalysts have been designed to function in alkaline environments. In most previous studies on electrode catalysts in anion-exchange membrane fuel cells (AEMFCs) or anion-exchange membrane water electrolyzers (AEMWEs), PGM-free transition metal-based electrocatalysts for oxygen reduction reaction (ORR) or oxygen evolution reaction (OER) have been widely studied because of their high potential for reducing the activation overpotentials of both ORR and OER, as well as their economic effectiveness [9]. Much effort has been devoted to the development of alternative materials, including transition-metal carbides, chalcogenides, phosphides, and nitrides. The non-noble metal properties and unique activity toward the hydrogen evolution reaction (HER) of transition metal-based phosphides, including FeP [10], CoP [11], $Ni_2P$ [12], MoP [13], and WP [14], have been extensively researched. Furthermore, Co-, Ni-, and Fe-based phosphides exhibit electrocatalytic activity toward the OER, in addition to their catalytic activity for the HER. Recent studies proved that CoP nanomaterials act as apt catalysts for the OER, which calls for the further investigation of this nanostructure [15]. However, most studies on CoP as electrocatalysts have been limited to AEMFCs or AEMWEs, and few reports on bifunctional catalysts have been published on their application in AEM-based URFCs.

In this study, noble-metal-free cobalt catalysts were prepared and used as bifunctional oxygen electrodes (BOE) for AEM-URFCs. For comparison, commercial Pt/C and IrB catalysts were used as bifunctional oxygen electrodes (BHE) and BOE, respectively, and tested in URFCs. The CoP catalyst showed better performance than that of $Co_3O_4$, which exhibits a current density of 145 mA cm$^{-2}$ at 0.1 V in fuel cell mode and 20 mA cm$^{-2}$ at 2.0 V in water electrolyzer mode. In addition, this improved the round-trip efficiency (RTE) compared to that of the $Co_3O_4$ catalyst. The CoP catalyst displayed better conductivity, which enhanced its catalytic performance. Electrochemical studies also supported the hypothesis that CoP has higher catalytic activity than the $Co_3O_4$ catalyst. Owing to the latest developments in AEM-URFC performance and cycling results, researchers have been able to construct PGM-free electrocatalysts for alkaline-based FC or WE.

## 2. Results and Discussion

The structural characterization of Co(OH)F, CoP, and $Co_3O_4$ was performed to evaluate the impact of the nanostructure on the URFC performance. The crystal structures of the synthesized Co(OH)F, CoP, and $Co_3O_4$ were characterized using XRD, as shown in Figure 1a. Co(OH)F exhibited major peaks at 20.8°, 35.6°, and 52.0° in the XRD patterns, which were assigned to the (110), (111), and (221) planes, respectively. It has an orthorhombic structure related to diaspore-type-AlOOH (SG: Pnma). The diffraction peaks are well matched with the standard pattern of Co(OH)F (JCPDS:50-0827) [16,17]. For CoP, an intense peak at 48.1° and peaks at 31.6°, 35.3°, 36.3°, and 46.2° can be observed in the XRD pattern (prepared at 450 °C), which indicates the diffraction plane (211), (011), (200), (111), and (112), respectively. In addition, a few other peaks with lower intensities at 52.2°, 56°, and 56.7° can be observed to the right of the 48.1° diffraction peak, corresponding to the (103), (020), and (301) diffraction planes, respectively. These diffraction peaks are assigned to the orthorhombic phase (JCPDS:20-0497) [16]. The obtained CoP has an orthorhombic Pnma space group. Figure S1 shows the XRD spectra of CoP calcinated at various other

temperatures, ranging from 300 to 400 °C. The presence of a peak at 20.8° indicated the presence of the parent precursor of Co(OH)F. The disappearance of this peak indicates the transition of CoP into a pure phase on calcinating the sample at 450 °C. Figure 1 also shows the XRD pattern of $Co_3O_4$, consisting of an intense peak at 2θ of 36.8°, indicating the (311) diffraction plane. The other peaks at 19°, 31.2°, 38.5°, 44.8°, 55.6°, 59.3°, and 65.2° observed on either side of the (311) plane are consistent with the diffraction planes (111), (220), (222), (400), (422), (511), and (440), respectively, as shown in JCPDS:42-1467 [17]. The presence of these diffraction peaks indicated the pure spinel nature of $Co_3O_4$. The crystallite size of CoP, Co(OH)F, and $Co_3O_4$ were calculated using the Debye–Scherrer equation. Co(OH)F and $Co_3O_4$ have higher crystallite sizes of 42 and 37.1 nm, respectively, when compared to CoP (21.4 nm). The smaller the size of the catalysts containing metal nanoparticles, the more enhanced the metal dispersion. The smaller crystallites of CoP disperse more evenly on the supporting matrix and augment the exposure of active sites, which leads to the efficient utilization of CoP catalyst and in turn enhances its catalytic activity [18].

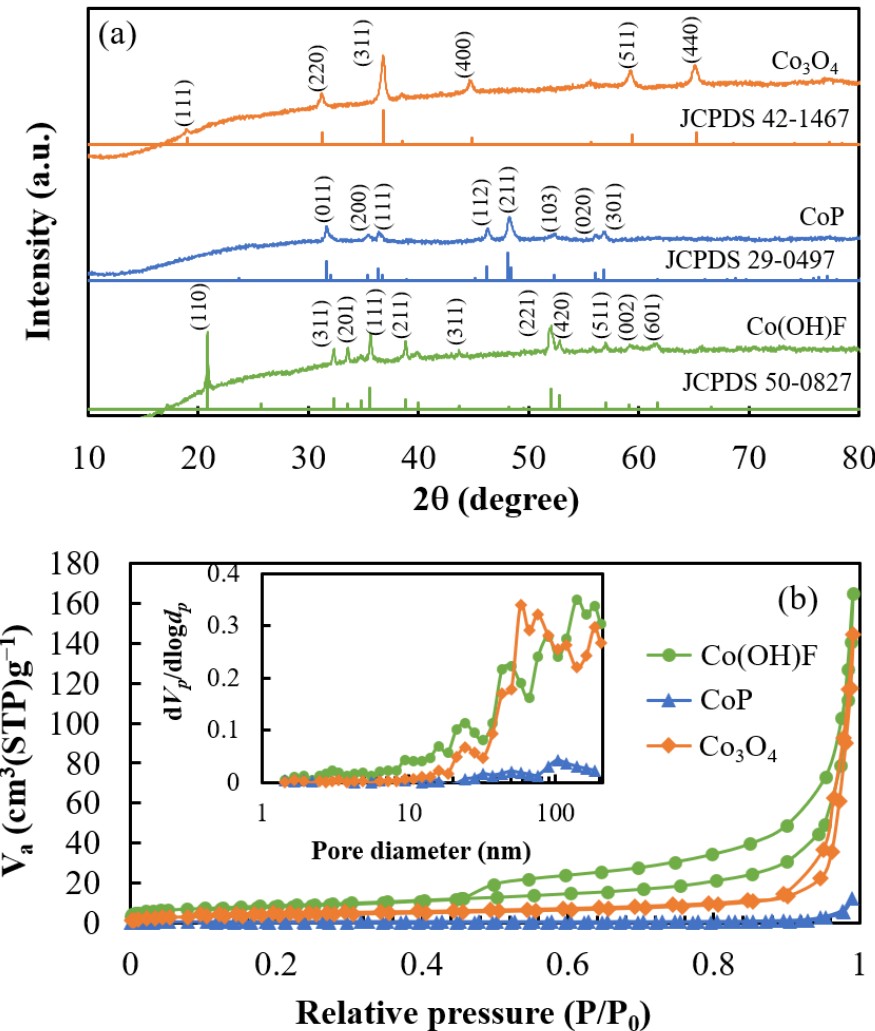

**Figure 1.** (**a**) XRD patterns and (**b**) $N_2$ adsorption/desorption isotherm curves of Co(OH)F, CoP, and $Co_3O_4$.

Figure 1b shows the $N_2$ adsorption–desorption isotherms of Co(OH)F, CoP, and $Co_3O_4$. The pore size distributions (Figure S2) show that all samples have mesopores and a type III isotherm curve, which indicates the possibility of multilayer adsorption owing to weak interactions between the adsorbed molecules and the presence of mesopores on the surface of the adsorbent [19]. When Co(OH)F was converted to CoP via phosphorization, the pore volume and surface area were decreased from 6.24 $m^2/g$ and 0.100 $cm^3/g$ to 2.46 $m^2/g$ and

0.019 cm³/g, respectively. However, those of $Co_3O_4$ (14.03 m²/g and 0.202 cm³/g) were significantly higher than those of Co(OH)F and CoP.

The surfaces and morphologies of the nanostructures are related to their water-splitting efficiencies [20]. FE-SEM and EDS images of the as-synthesized Co(OH)F, CoP, and $Co_3O_4$ are depicted in Figure 2. Co(OH)F and CoP exhibited a clear rod-like morphology, as shown in Figure 2a,c. The formation of rods may be attributed to the addition of $NH_4F$ [21,22]. The surface of the CoP became smooth owing to the phosphorization of Co(OH)F, which induced a decrease in the surface area. Figure 2e shows the morphology of $Co_3O_4$ to consist of a mixture of flakes and rods. The disruption of the rod morphology can be attributed to the calcination in air. The pores and nano protrusion in CoP are marked as circles and squares in Figure 2c. The elemental mapping of these materials via EDS indicates the presence of Co, F, P, and O, as shown in Figure 2b,d,f.

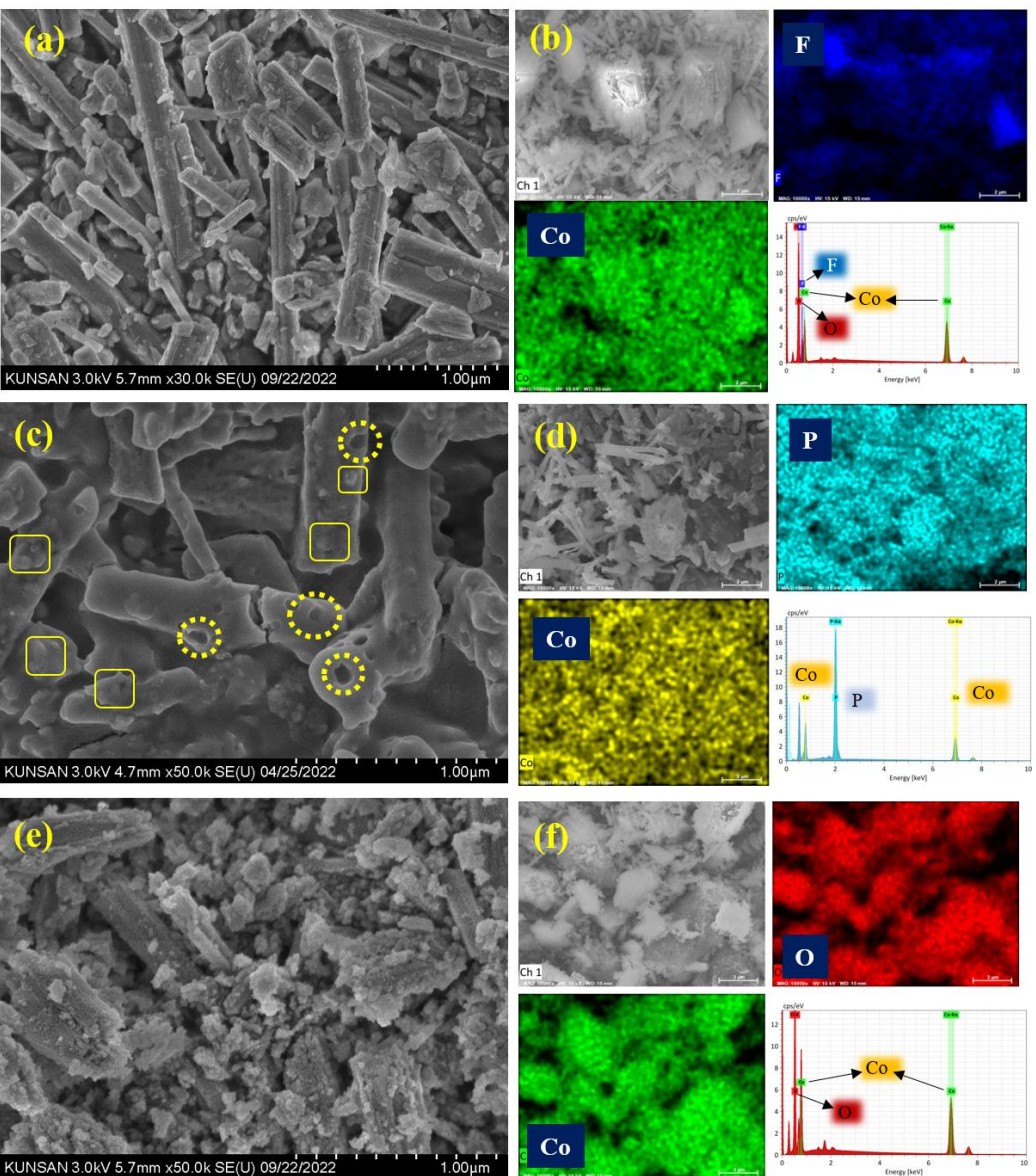

**Figure 2.** SEM images EDS mapping and spectra of (**a,b**) Co(OH)F, (**c,d**) CoP (Yellow dotted circle: Pores, Squares: Nano protrusion), and (**e,f**) $Co_3O_4$.

The electrocatalysis of catalysts depends on their intrinsic properties, size, shape, and the functionalization of the nanomaterial. The rod-like morphology of the nanostructures exposed more active sites and facilitated electron transfer. The TEM images of Co(OH)F,

CoP, and $Co_3O_4$ in Figure 3 clearly show the rod shape structures. Unlike Co(OH)F and CoP, $Co_3O_4$ had the presence of disconnected spherical structures that formed very fine rod-like structures. The nano protrusions seen on CoP enhance its electrocatalytic activity owing to the presence of active sites that aid electron transportation. Thus, the rod-shaped structure of the materials has the advantages of a mass diffusion pathway and an active surface area; hence, it is expected to be a suitable surface [23]. The crystalline fringe patterns of Co(OH)F, CoP, and $Co_3O_4$ in the insets of Figure 3 show that these materials were well crystallized. The fringe patterns of Co(OH)F, CoP, and $Co_3O_4$ with lattice spacings of 0.25, 0.19, and 0.24 nm are associated with the (111), (211), and (311) diffraction planes, respectively.

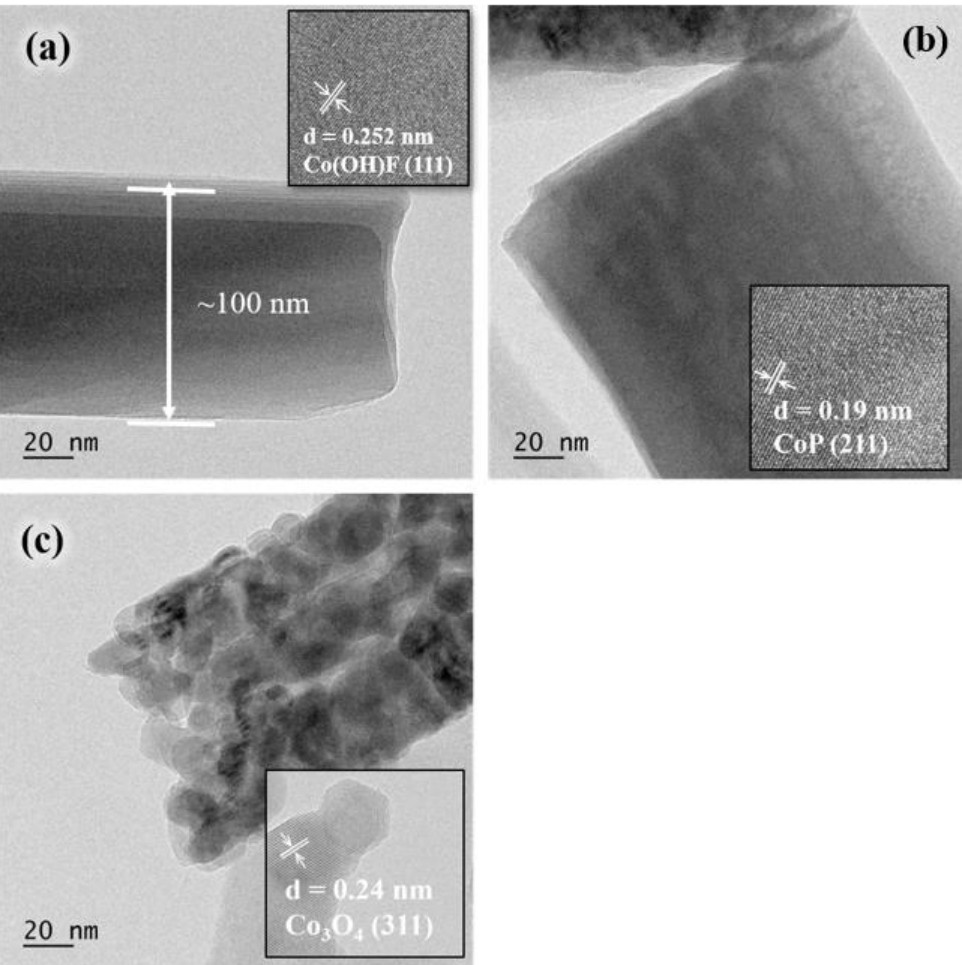

**Figure 3.** TEM images of (**a**) Co(OH)F, (**b**) CoP, and (**c**) $Co_3O_4$.

The combination of elements in the catalysts was investigated using XPS, as shown in Figure 4 and Figure S3. The survey spectra in Figure 4a show the presence of Co, F, P, and O peaks, indicating the successful formation of Co(OH)F, CoP, and $Co_3O_4$. These findings were further confirmed through the presence of these elements in the EDX spectra, as discussed earlier. Figure 4b shows the deconvoluted Co 2p spectrum. The Co 2p spectrum is shown in Figure 4b. Co(OH)F had 777.6, 782.1, and 793.7 assigned to $Co^{3+}$ and $Co^{2+}$ of Co $2p_{3/2}$, and $Co^{2+}$ of Co $2p_{1/2}$, respectively. This indicates that most cobalt ions in Co(OH)F exist as $Co^{2+}$. CoP shows peaks at 778.1 and 794.7 eV, corresponding to Co $2p_{3/2}$ and Co $2p_{1/2}$, which in turn correspond to the Co–P bond [19,24]. The peaks at 772, 783, and 800 eV are attributed to satellite peaks [25]. The spectra of Co 2p in $Co_3O_4$ also had the two major peaks at 777.8 and 792.8 eV assigned to Co $2p_{3/2}$ and Co $2p_{1/2}$, respectively. The subpeaks obtained through the curve fitting of the Co $2p_{3/2}$ peak can be attributed to the $Co^{2+}$ (779.1 eV) and $Co^{3+}$ (776.9 eV) valence states. Similarly, the Co $2p_{1/2}$ peak consists

of two sub-peaks at 792.04 eV and 794.3 eV, corresponding to the $Co^{3+}$ and $Co^{2+}$ valance states [26].

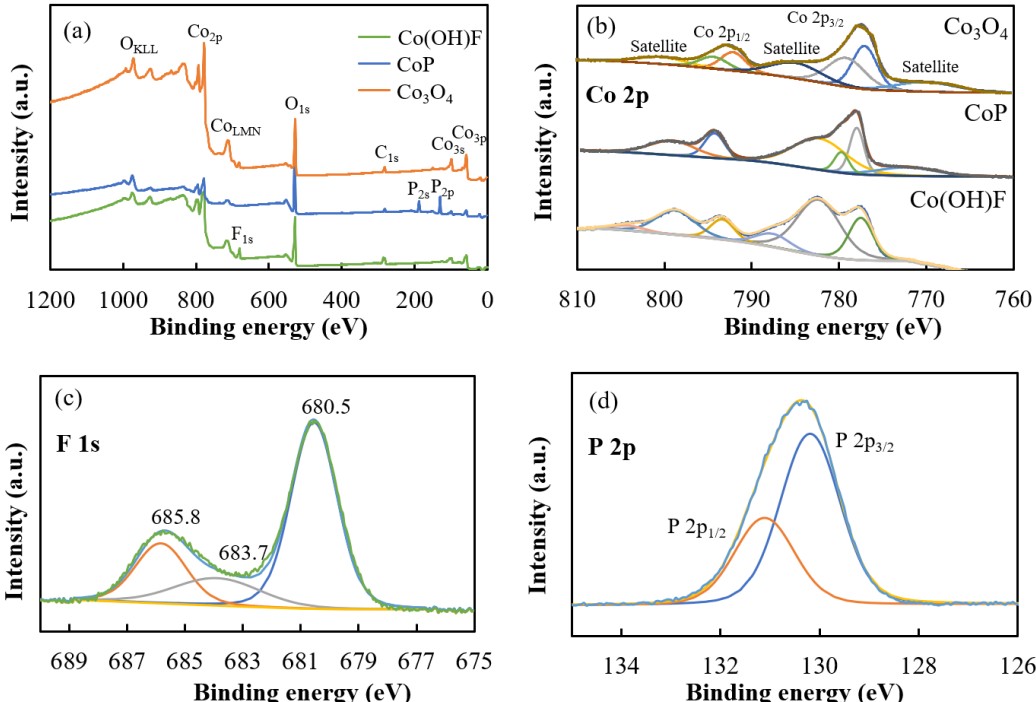

**Figure 4.** XPS spectra of Co(OH)F, CoP, and $Co_3O_4$. (**a**) Overview, (**b**) Co 2p, (**c**) F 1s of Co(OH)F, and (**d**) P 2p of CoP.

Figure 4c,d show peaks at 680.5 and 685.8, and 130 and 131 eV, indicating the presence of F 1s on Co(OH)F and P $2p_{3/2}$ and P $2p_{1/2}$ in CoP. The binding energy of Co 2p shows a positive shift and that of P 2p exhibits a negative shift, which is evidence of the transfer of electrons from cobalt to phosphorous, thus making Co positively and P negatively charged in CoP, confirming the formation of the same [27]. From Figure S4, it is evident that the deconvoluted spectra of O 1s for Co(OH)F and $Co_3O_4$ consist of peaks at 526.2 eV and 527.7 eV, which can be attributed to Co–O, and the peak at 533.5 eV corresponds to the oxygen vacancies. The adsorption energy of water is considerably reduced owing to the presence of oxygen vacancies. The presence of oxygen vacancies may improve the catalytic performance of $Co_3O_4$ catalysts in the OER [27]. Additionally, the presence of peak at 529.3 eV indicates the binding energy of oxygen isotope $^{16}O$. The presence of this isotope provides evidence supporting the participation of lattice oxygen, present in catalysts in the OER [28].

The MEA was prepared using the CCM method. It is possible to reduce MEA resistance using the CCM method, which decreases the membrane/catalyst layer interface. The Co catalyst was coated on one side and Pt/C was coated on the other side of the membrane, and these were used as the oxygen and hydrogen electrodes, respectively. A PGM-free electrocatalyst with bifunctional ORR and OER activity is preferable, replacing the expensive Ir–B as the BOE. In order to perform the cell test, the membrane was immersed in KOH to change the $OH^-$ form. Figure 5a shows the polarization curves of the cell performance in the URFCs (fuel cell and water electrolyzer modes). The Co(OH)F, CoP, and $Co_3O_4$ catalysts delivered current densities of 77.5, 105.0, and 62.5 mA $cm^{-2}$ at 0.3 V in the fuel cell mode. The fuel cell performances confirm that the CoP catalyst performs better than Co(OH)F and $Co_3O_4$. The performance is comparable to that of the commercial Ir–B samples (117.5 mA $cm^{-2}$ at 0.3V). The higher electron-donating ability of P can lead to the high distortion of the surface charge of the catalyst, favoring the adsorption of oxygen intermediaries [29]. In particular, CoP has a strong electron capture capability that promotes

ORR kinetics [30–32]. The electron transfer ability and conductivity of CoP are improved via the interpenetration of the O–P bond, which occurs due to the doping effect, enhancing its ORR property [33]. However, the order of water electrolyzer performance was Co(OH)F > $Co_3O_4$ > CoP, which had current densities of 30, 25, and 20 mA cm$^{-2}$ at 2.0 V, respectively. The water electrolysis performances in this work were relatively low compared to the results reported in the literature because of the supply of pure water not KOH solution. The current of Co(OH)F, which was higher than that of CoP, may be induced via the surface area and original catalytic activity for the oxygen evolution reaction [34,35]. It has been reported that CoP is more active compared with $Co_3O_4$ for the OER in alkaline solutions because anionic vacancies decrease the activation energy of the OER, and P vacancies act as active sites toward the OER [36–38]. In this work, the high performance of the catalysts for the OER results mainly from increasing the density of active sites (large surface areas) rather than intrinsic catalytic activity. Figure 4b shows the Tafel plots for the fuel cell performance. CoP displayed a Tafel slope (74.5 mV/dec) similar to that shown by Ir–B and much lower than those of Co(OH)F (82.7 mV/dec) and $Co_3O_4$ (88.9 mV/dec). The wide acceptance of the Tafel slopes of 120, 40, and 30 mV/dec for the Volmer, Heyrovsky, and Tafel determining rate steps, respectively, serves as validation for our kinetic model [39]. The Tafel slope values of the prepared catalysts, ranging from 75 to 90 mV/dec, further confirm their adherence to the Volmer–Heyrovsky reaction pathway [40,41]. The highest power density of the fuel cell using CoP was 32.375 mW cm$^{-2}$, which was higher than those of Co(OH)F (23.625 mW cm$^{-2}$) and $Co_3O_4$ (20.125 mW cm$^{-2}$), as shown in Figure S4a. In addition, the prepared catalysts were compared with previously reported BOE catalysts (Table 1). RTE measures the difference between the voltages of the fuel cell and water electrolyzer mode to observe the constant current. The CoP catalyst shown in Figure S4b shows an RTE of 30.0%, which is similar to those of the Co(OH)F (30.6%) and $Co_3O_4$ (28.7%) catalysts. The results confirm that the CoP has better performance, particularly in fuel cell mode, compared to other non-noble metal catalysts.

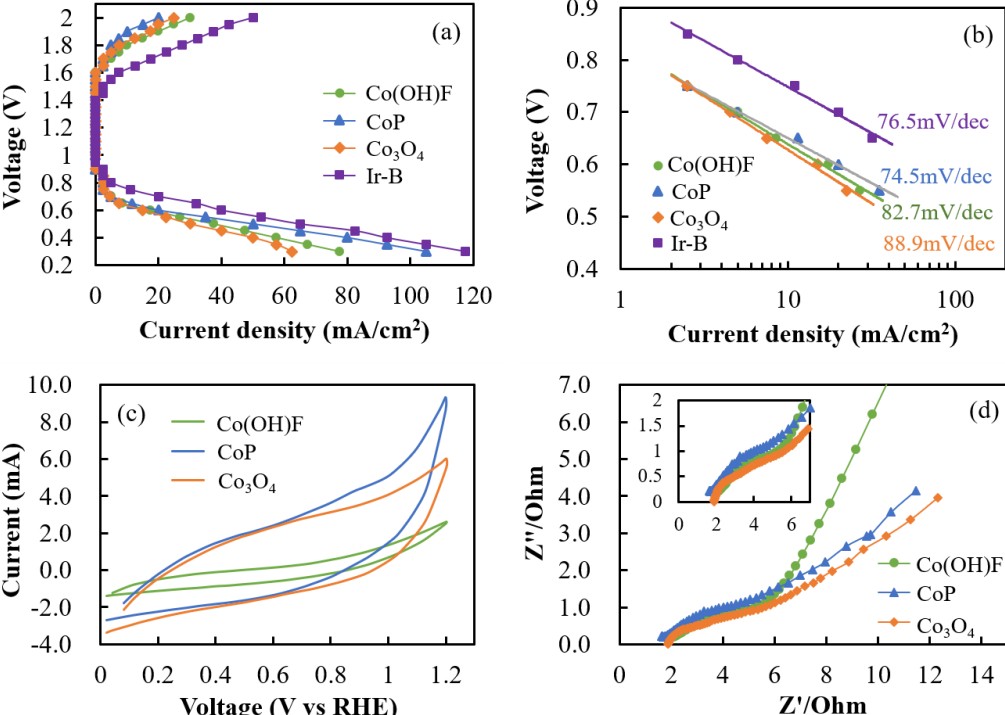

**Figure 5.** (**a**) IV curves, (**b**) Tafel plots, (**c**) cyclic voltammograms, and (**d**) electrochemical impedance spectra of Co(OH)F, CoP, $Co_3O_4$, and IrB catalysts as BOE with Pt/C as BHE.

In order to clarify the electrochemical mechanism underlying the high performance of the Co(OH)F, CoP, and $Co_3O_4$ catalysts, cyclic voltammetry and electrochemical impedance

spectroscopy (EIS) were used, and results are shown in Figures 5c,d and S5. Through cyclic voltammetry (CV), the electrochemically effective double-layer capacitance ($C_{dl}$) was evaluated and the active surface areas of the catalysts was determined. As shown in Figure 5c, CoP delivers a higher current response and double-layer capacitance than the Co(OH)F and $Co_3O_4$ catalysts, suggesting that more active sites are introduced in the CoP catalyst, resulting in improved catalytic performance. To evaluate the stability of the catalysts, we conducted 50 cycles of cyclic voltammetry (CV) for the prepared catalysts, as illustrated in Figure S6. The CV curve after 50 cycles demonstrates that the CoP catalyst exhibits a higher current response compared to the $Co_3O_4$ and Co(OH)F catalysts. This result confirms that CoP exhibits superior catalytic behavior compared to the other catalysts studied. From the EIS measurements shown in Figure 5d, the CoP catalyst (3.1 Ω) has a lower charge-transfer resistance than the Co(OH)F (3.22 Ω) and $Co_3O_4$ catalysts (3.63 Ω), suggesting that CoP is capable of faster charge transfer than Co(OH)F and $Co_3O_4$. This may be because CoP's intrinsic metallic characteristics produce good electrical contact between the catalyst and its support, resulting in the rapid electron transfer between the electrode and the catalyst [42]. By analyzing the Nyquist plot, the calculated values for Rs are found to be 1.6 Ω for CoP, 1.8 Ω for $Co_3O_4$, and 2.1 Ω for Co(OH)F. It can be concluded that the Rs value for CoP is lower compared to $Co_3O_4$ and Co(OH)F. Lower Rs values indicate lower solution resistance, which suggests a more conductive electrolyte and better charge transfer kinetics in the electrochemical system [43].

In this work, the electrochemical studies of the cell performance, CV, and EIS showed the catalytic activities of Co(OH)F, CoP, and $Co_3O_4$ nanorods as BOE catalysts. CoP showed fuel cell performance comparable to that of Ir–B, and Co(OH)F exhibits the highest water electrolysis performance when compared to CoP and $Co_3O_4$. Thus, Co(OH)F and CoP can be used as catalysts for URFC. The catalysts prepared in this study followed fluorination and phosphorization strategies, which accelerated electron transport and charge transfer. In addition, the phosphorization reaction introduces oxygen vacancies, which modulate the electronic structure and provides active sites on CoP, thereby exhibiting better electrocatalytic performances [44].

**Table 1.** Comparison for the electrochemical performance of prepared catalysts with previously reported results.

| Catalyst (mg cm$^{-2}$) | | Membrane | FC | WE | Ref. |
|---|---|---|---|---|---|
| **BHE** | **BOE** | | | | |
| Pt/C (2.0) | Co(OH)F (2.0) | FAA-3-50 | 77 (0.3 V) | 30 (2.0 V) | This work |
| Pt/C (2.0) | CoP (2.0) | FAA-3-50 | 105 (0.3 V) | 20 (2.0 V) | |
| Pt/C (2.0) | $Co_3O_4$ (2.0) | FAA-3-50 | 62 (0.3 V) | 25 (2.0 V) | |
| Pt/C (2.0) | Ir-black (2.0) | FAA-3-50 | 117 (0.3 V) | 50 (2.0 V) | |
| Pt/C (0.5) | $MnO_x$-SS (0.3) | FAA-3-PK-130 | 65 (0.4 V) | 58 (1.7 V) | [45] |
| Pt/C (0.5) | $MnO_x$/Ni-CP (0.3) | FAA-3-PK-130 | 65 (0.4 V) | 20 (1.8 V) | |
| Pt/C (0.5) | Fe-N-C + NiFe-LDH/C (0.5) | A201 | 95 (0.3 V) | 90 (1.8 V) | [46] |
| Ni/C (6.0) | Ni/C + $MnO_x$/GC (4.0) | FAA3 | 24 (0.4 V) | 17 (1.8 V) | [47] |

## 3. Materials and Methods

For the preparation of CoP nanorods, 1 mmol (0.291 g) of cobalt (II) nitrate hexahydrate ($Co(NO_3)_2$ $6H_2O$, MW 291.03) was dissolved in 36 mL of DI water along with 8 mmol (0.296 g) of ammonium fluoride ($NH_4F$, MW 37.04) and 10 mmol (0.6 g) of urea ($CO(NH_2)_2$, MW 60.06). The solution was then stirred for 30 min and transferred to a Teflon-lined autoclave for heat treatment at 120 °C for 8 h. The obtained product was washed thoroughly with DI water and dried at 50 °C for 12 h under vacuum to obtain Co(OH)F. Then, Co(OH)F and $NaH_2PO_2$ at a mass ratio of 1:10 were added to two separate porcelain boats. Initially, Ar gas (50 sccm) was passed upstream from the $NaH_2PO_2$ and followed by heating the samples to 450 °C at the rate of 5 °C min$^{-1}$ and maintained for 2 h. The furnace was then

cooled down to room temperature to obtain CoP nanorods. A schematic representation of the preparation procedure is shown in Figure S7. To prepare $Co_3O_4$, Co(OH)F was placed in a porcelain boat and heated in air without $NaH_2PO_2$.

Powder X-ray diffraction (XRD; SmartLab SE, Rigaku, Tokyo, Japan) was used to determine the crystal structures of the samples. Scanning electron microscopy (SEM, SU8220, Tokyo, Japan) and transmission electron microscopy (TEM, JEM-2100 LaB6, USA) were used to characterize the morphologies. The elemental composition was determined using energy-dispersive X-ray spectroscopy (EDX; SU8220, Tokyo, Japan). The electronic states of the samples were determined via X-ray photoelectron spectroscopy (XPS, Axis Supra+, Kratos Analytical, Manchester, UK). The surface area and porosity were measured using $N_2$ adsorption/desorption analysis (Brunauer–Emmett–Teller, BELSORP-max, Osaka, Japan).

For the preparation of the membrane electrode assembly (MEA) for the cell test, 70% the of catalyst (2 mg cm$^{-2}$) was placed in a glass bottle to which 2 mL of distilled water (DW) and 3 mL of Isopropyl alcohol was added, followed by the addition of 30% of ionomer (FAA-3-SOULT-10, Fumatech, Bietigheim-Bissingen, Germany) into this solution. The solution was homogenized via ultrasonication before coating the membrane using a spray gun maintained at 60 °C. The other side of the membrane was coated with Pt/C as a catalyst in a BHE containing 70% Pt and 30% ionomer. Subsequently, the catalyst-coated membrane (CCM) was immersed in DW for 30 min, transferred to a solution containing 1 M KOH, and left isolated. This membrane was left undisturbed overnight and immersed in DW for 1 h prior to application to the device. The schematic illustration of the procedure is given in Scheme S1.

The URFC test procedure and conditions are described in a previous study [48]. An AEM-URFC test station with a single cell was set up. As the cells were started up, humidified nitrogen was purged into the anode and cathode at a rate of 100 cc min$^{-1}$ for 30 min. The humidifiers were set to 50 °C for the oxygen and hydrogen sides. After all set temperatures were stabilized, the nitrogen flow was halted. Then, fully humidified hydrogen and oxygen (99.99% pure) gases were passed through the BHE and BOE sides as an initial step for the performance evaluation of the AEM-URFC. Hydrogen and oxygen flow rates were regulated at 100 and 200 cc/min, respectively. The cell temperature was maintained at 50 °C prior to the activation. The cell was then operated at 0.1 V for 3 h, and the cell polarization curve for the FC mode was recorded from an open-circuit voltage of 0.1 V. In order to evaluate the performance of the cell in WE mode, pure $H_2O$ at a rate of 2.25 mL min$^{-1}$ from a water reservoir was supplied to both electrodes. The cell was subjected to activation for 3 h at 2.0 V. Then, the polarization curve was recorded from 1.5 V to 2.0 V. A power supply (MK POWER, MK 3010P, Seoul, Korea) was used to operate the device in the WE mode, and an electric loader (DAE GIL, EL-200P, Seoul, Korea) was used to evaluate the cell performance in the FC mode.

## 4. Conclusions

Different types of cobalt-based nanorod catalysts were examined in the BOE of AEM-URFCs. Fluorine and phosphorous were successfully incorporated into the catalyst via fluorination and phosphorization, and Co(OH)F and CoP nanorods were synthesized using hydrothermal and heat treatments, as confirmed through XRD and XPS. Crystallized nanorods with ~100 nm diameter were observed in the TEM images. In the AEM-URFC test, the CoP catalyst in the BOE delivered the best performance in the fuel cell mode (105 mA cm$^{-2}$ at 0.3 V), and Co(OH)F displayed a good performance in the water electrolyzer mode (30 mA cm$^{-2}$ at 2.0 V). CoP and Co(OH)F exhibited higher RTE and power densities than the conventional $Co_3O_4$ catalyst, although $Co_3O_4$ had a relatively high surface area. This may be because of its high conductivity and intrinsic catalytic activity. Owing to these promising results, AEM-URFC applications can be realized using noble-metal-free Co-based catalysts such as BOE.

**Supplementary Materials:** The following supporting information can be downloaded at: https: //www.mdpi.com/article/10.3390/catal13060941/s1, Figure S1: XRD patterns of CoP at different temperatures; Figure S2: Pore size distribution curves of Co(OH)F, CoP, and $Co_3O_4$; Figure S3: Deconvoluted O1s spectra of (a) $Co_3O_4$ and (b) Co(OH)F samples; Figure S4. (a) Power density and (b) RTE of Co(OH)F, CoP, and $Co_3O_4$ for fuel cell performance; Figure S5. After fuel cell mode (a) CV and (b) EIS of IrB sample as the oxygen electrode and Pt/C as the hydrogen electrode; Figure S6. 50th cycle CV curves of Co(OH)F, CoP, and $Co_3O_4$ catalysts; Figure S7: Schematic diagram of the preparation procedure; Scheme S1: Schematic illustration of MEA preparation and cell assembly.

**Author Contributions:** P.R.: conceptualization, methodology, data curation, writing—original draft. M.M.R.: methodology, data curation. B.-S.K.: data curation. H.-J.S.: formal analysis. G.P.: investigation. S.-Y.K.: formal analysis. H.-K.L.: investigation. J.S.: writing—review and editing, supervision, validation, project administration, funding acquisition. All authors have read and agreed to the published version of the manuscript.

**Funding:** This work was supported by the National Research Foundation of Korea (NRF), grant funded by the Korean government (MSIT) (No. 2021R1I1A3057906); the Korea Institute of Energy Technology Evaluation and Planning (KETEP), grant funded by the Korean government (MOTIE) (20224000000220, Jeonbuk Regional Energy Cluster Training of human resources); and the Human Resources Program for the EV industrial cluster, Gunsan City, Korea.

**Data Availability Statement:** Not applicable.

**Conflicts of Interest:** The authors declare no conflict of interest.

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
