# Peer review of "Precious Metal-Free CoP Nanorod Electrocatalyst as an Effective Bifunctional Oxygen Electrode for Anion Exchange Membrane-Unitized Regenerative Fuel Cells"

_catalysts, doi:10.3390/catal13060941_

Round 1
Reviewer 1 Report
In this work, the authors compared the performance of Co(OH)F, CoP and Co3O4 in anion exchange membarance unitized fuel cells and electrolyzers. The utilization of catalysts in real applications is important and this manuscript can be accepted after well solving my comments and suggestions.
1. Many format mistakes exist in the text and figures, such as superscripts, subscripts, abbreviations, unit format, and spelling. The authors should check and correct these mistakes through the manuscript. Also, the English should be further improved.
2. Necessary references should be given about the JCPDS cards to support the structural information.
3. Some words in Fig. 3a are too small and not clear.
4. Please explain why O 1s signals also exist in the CoP sample (Fig. 4a).
5. The analysis of O 1s XPS is not clear, please refer to this paper with DOI of 10.1002/adfm.202207618 about the correct analysis on O 1s XPS.
6. The semicircle of the EIS plots should be analyzed in Fig. 5d. Please refer to this paper with DOI of 10.1063/5.0083059.
7. The stability tests should be conducted and compared.
Minor editing of English language required.
Reviewer 2 Report
Reviewer’s comments
1. Line 21-22 reads, “It was synthesized from Co(OH)F by hydrothermal treatment 21 of cobalt nitrate, ammonium fluoride and urea, and its phosphorization.” This statement is unclear, what does “it” refers to? I suggest the authors use the name instead of “it”, and ensure that the statement is clear.
2. There are grammatical problems in the following: line 49 “Further”, line 59 “high promise”, line 79 “contrive”. In addition to these issues, there are other grammatical issues such as the omission of using articles “the” in most statements. Thus I suggest the authors take the paper for English editing.
3. Line 72 reads, “For comparison the commercial Pt/C and IrB catalysts”, usually IrO or RuO are used as commercial catalysts for Oxygen evolution/reduction. Thus, the authors should explain the IrB catalyst.
4. 2θ is missing in line 97.
5. The authors should calculate the particle sizes of Co(OH)F, CoP and Co3O4 using the Debye-Scherrer equation from the results in figure 1a. Afterwards, they should relay the impact of the particle size on the catalytic activity on their discussion section.
6. The authors should redraw figure 1b and change the scale to accommodate the CoP graph, so that it can be clear to enable readser to visualize the type of the isotherm from the graph.
7. “Figure 1(b) show the pore size distributions and N2 adsorption-desorption isotherms of Co(OH)F, CoP and Co3O4. The pore size distributions show all samples had macropores and depicted type III isotherm curve, which indicate the possibility of multilayer adsorption due to weak interactions among adsorbed molecules and the presence of macropores on the surface of adsorbent [23].” The authors made this statement, however, the graphs suggest otherwise. The graphs show a hysteresis loop that is a key characteristic of mesoporous materials. The authors should find an explanation for this, otherwise they should reanalyse their materials. Furthermore, figure CoP seems to be different than the other twpo materials, but the authors suggest it is also a type III isotherm, thus they should revisit this section.
8. “When Co(OH)F was converted to CoP by phosphorization, the pore volume and surface area were decreased from 6.24 m2/g and 0.100 cm3/g to 2.46 m2/g and 0.019 cm3/g, respectively. However, those of Co3O4 (14.03 m2/g and 0.202 cm3/g) were significantly higher than Co(OH)F and CoP”. The surface area values are too low for CoP and since surface reactions are expected, why would the authors convince anyone to use this materials and benefit from their best performance? It makes one wonder if this material is working to its best performance or rather it is operating at its lowest?
9. Line 125 reads “The electrocatalysis of catalysts depend on their own nature and physical surface area.” This statement doesn’t correlate with what the nitrogen adsorption-desorption results are showing. The authors need to explain this.
10. Was the EDX and XPS in agreement? The authors should relate the two in the discussion.
11. Can the authors analyse the rate determining step from the Tafel plots obtained, the values of the Tafel plots obtained should tell then more about the rate of reactions, thus they can estimate what is happening in each reaction step.
12. Nothing was said about the solution resistance (Rs) and the Warburg resistance (W) from the EIS, can the authors explain the influence of these two resistance types on the performance of their catalysts.
13. How stable were the catalysts? The authors should perform studies on assessing the long term stability of the catalysts.
Look at comment 2 on the reviewer report.
Round 2
Reviewer 2 Report
The authors have successfully addressed all comments.